# *Verticillium dahliae* VdTHI20, Involved in Pyrimidine Biosynthesis, Is Required for DNA Repair Functions and Pathogenicity

**DOI:** 10.3390/ijms21041378

**Published:** 2020-02-18

**Authors:** Tengfei Qin, Wei Hao, Runrun Sun, Yuqing Li, Yuanyuan Wang, Chunyan Wei, Tao Dong, Bingjie Wu, Na Dong, Weipeng Wang, Jialiang Sun, Qiuyue Yang, Yaxin Zhang, Song Yang, Qinglian Wang

**Affiliations:** 1Henan Collaborative Innovation Center of Modern Biological Breeding, Henan Institute of Sciences and Technology, Xinxiang 453003, China; qintengfeisam@gmail.com (T.Q.); sunrunrun123@gmail.com (R.S.); lyq120327@gmail.com (Y.L.); wangyy1007@gmail.com (Y.W.); chunyanwei688@gmail.com (C.W.); dongnana1229@gmail.com (N.D.); wweipeng125@gmail.com (W.W.); swjl5845@gmail.com (J.S.); yaxinzhang33@gmail.com (Y.Z.); yangsong11223@gmail.com (S.Y.); 2School of Life Sciences, The Chinese University of Hong Kong, Hong Kong 999077, China; haowei9111@gmail.com; 3College of Agriculture, Liaocheng University, Liaocheng 252059, China; wubingjie@lcu.edu.cn

**Keywords:** *Verticillium dahliae*, THI20, thiamine biosynthesis, pathogenicity

## Abstract

*Verticillium dahliae* (*V. dahliae*) infects roots and colonizes the vascular vessels of host plants, significantly reducing the economic yield of cotton and other crops. In this study, the protein VdTHI20, which is involved in the thiamine biosynthesis pathway, was characterized by knocking out the corresponding *VdTHI20* gene in *V. dahliae* via *Agrobacterium tumefaciens*-mediated transformation (ATMT). The deletion of *VdTHI20* resulted in several phenotypic defects in vegetative growth and conidiation and in impaired virulence in tobacco seedlings. We show that VdTHI20 increases the tolerance of *V. dahliae* to UV damage. The impaired vegetative growth of *ΔVdTHI20* mutant strains was restored by complementation with a functional copy of the *VdTHI20* gene or by supplementation with additional thiamine. Furthermore, the root infection and colonization of the *ΔVdTHI20* mutant strains were suppressed, as indicated by green fluorescent protein (GFP)-labelling under microscope observation. When the RNAi constructs of *VdTHI20* were used to transform *Nicotiana benthamiana*, the transgenic lines expressing dsVdTHI20 showed elevated resistance to *V. dahliae*. Together, these results suggest that VdTHI20 plays a significant role in the pathogenicity of *V. dahliae*. In addition, the pathogenesis-related gene *VdTHI20* exhibits potential for controlling *V. dahliae* in important crops.

## 1. Introduction

The soilborne phytopathogen *Verticillium dahliae* causes vascular wilt diseases, which devastatingly attacks many important crops [1,2]. A severe loss in the yield of cotton results from verticillium wilt in China every year, causing severe economic losses in the cotton industry [3,4]. Until very recently, no effective fungicide could cure this damaging disease because the hyphae of *V. dahliae* spread inside the xylem tissues, where they cannot be reached by fungicides [5]. The molecular mechanisms underlying the pathogenesis of this fungus remain unknown. Thus, studying growth- and virulence-related genes is of great importance for controlling this fungus.

*V. dahliae* can remain dormant in soil and dead plants for many years and invade the roots of nearby host plants. Hyphae adhere to and penetrate plant roots and then grow into the vascular system. With abundant budding and growth, the transportation of nutrients in the host is prevented, leading to various symptoms, including stunting, vascular browning, wilting, and foliar chlorosis and necrosis. Finally, the fungus completes its life cycle after its release into the soil [6]. Thiamine, or water-soluble vitamin B, a vital metabolic enzyme, acts as a cofactor in amino acid and carbohydrate metabolism in all living organisms [7]. Only microorganisms and plants can synthesize thiamine naturally, whereas animals cannot. Thiamine also plays a vital role in different fungal life cycles, such as growth, development, sporulation and host invasion.

Thiamine consists of two aromatic components, a pyrimidine (2-methyl-4-amino-5-hydroxymethylpyrimidine, HMP) and a thiazole (4-methyl-5-β-hydroxyethylthiazole, HET), which are synthesized in independent branches of the thiamine synthesis pathway. Thiamine pyrophosphate, phosphorylated thiamine (TPP), is the predominant active form of thiamine [8,9]. In *Saccharomyces cerevisiae,* many genes were found to be involved in thiamine pathway. *THI5*, *THI11*, *THI12,* and *THI13* are *THI5* gene family members that their function is redundant in HMP-P synthesis [10]. Three members of the THI20 family proteins, THI20/21/22, were characterized in *S. cerevisiae* [11]. *THI20* and *THI21* are two redundant genes that encode the HMP-P kinase of *S. cerevisiae*. THI20/THI21 can not only phosphorylate HMP to HMP-P but also phosphorylate HMP-P to HMP-PP [12]. Moreover, THI20 has been indicated to be a trifunctional protein with both thiamine biosynthetic and degradative activity [13]. THI4, involved in HET-P synthesis in *S. cerevisiae*, could play a role in DNA damage tolerance [14]. THI6 has been showed to have the thiamin-phosphate pyrophosphorylase and hydroxyethylthiazole kinase activity [15]. *THI80* is a thiamin-pyrophosphokinase gene that catalyzes thiamine to the active form (TPP) [16]. *THI10*, which encodes a thiamine transport protein that transports the thiamine from periplasm to the cell [17]. PHO3 is a periplasmic acid phosphatase that allows thiamin phosphates hydrolysis in periplasm [18]. In recent years, several genes in the thiamine biosynthesis pathway have been found in *V. dahliae*. VdTHI4, involved in HET-P synthesis, is required for fungal induction in tomato, as demonstrated by the infection analysis of the knockout strain *∆VdTHI4* [19]. VdThit, a thiamine transport protein, is important for the pathogenicity of *V. dahliae*, as supported by infection assays with the *V. dahliae VdΔTHit* deletion strain (Table 1) [20]. HET biosynthesis and thiamine transport for the thiamine biosynthesis pathway seem to influence the virulence and pathogenesis of *V. dahlia*. Only one THI20 family gene, *VdTHI20*, was found in *V. dahliae* (Figure 1). However, the mechanism by which *VdTHI20* contributes to the virulence and pathogenesis of *V. dahliae* has not been reported.

In the present study, we generated a targeted *VdTHI20* (VDAG_05690) mutant, *ΔVdTHI20*. We characterized its growth, development, stress tolerance, penetration ability and virulence in *Nicotiana benthamiana*. Decreased colony growth was detected as a result of the deletion of *VdTHI20*. The virulence of the mutants was significantly decreased compared with that of the wild-type and complemented strains. The germination of *ΔVdTHI20* was significantly impaired, as determined by confocal microscopy observations. Furthermore, transgenic *N. benthamiana* expressing dsVdTHI20 exhibited resistance to *V. dahliae.* Overall, our results indicate that *VdTHI20* plays a critical role in the pathogenicity of *V. dahliae*.

## 2. Results

### 2.1. Deletion and Complementation of VdTHI20 in V. dahliae

The *VdTHI20* gene was replaced with a geneticin-resistance cassette (*neo*) via homologous recombination in *Vd-wt*. Neomycin-resistant transformants were verified by genomic PCR. Physical maps of the *VdTHI20* locus and of the homologous recombination constructs were obtained by fusion of the *Vd**THI20* up flank, a *neo* resistance gene cassette and *Vd**THI20* down flank (Figure 2A). The *VdTHI20* gene was successfully replaced in 6 out of 210 analysed transformants. Two *ΔVdTHI20* mutants (*ΔVdTHI20-2* and *ΔVdTHI20-6*) were selected for further analysis (Figure 2B). Moreover, a functional copy of *VdTHI20* was introduced into *ΔVdTHI20-2* and *ΔVdTHI20-6* for complementation, resulting in two complemented strains, *ΔVdTHI20-2-c* and *ΔVdTHI20-6-c*. Successful transformation was confirmed by genomic PCR with the primers hyg-F/hyg-R and THI20-F/THI20-R (Figure 2C). *ΔVd**THI20*-*2*-*C* and *ΔVd**THI20*-*6*-*C* were selected for further phenotypic observations.

### 2.2. Radial Mycelial Growth in ∆VdTHI20 Mutants Was Significantly Reduced

The function of *VdTHI20* was investigated by growing *Vd-wt*, two mutants (*ΔVdTHI20-2* and *ΔVdTHI20-6*) and two complemented stains (*ΔVdTHI20-2-c* and *ΔVdTHI20-6-c*) on MM agar with different carbon sources (pectin, sucrose, galactose, xylose or starch). The growth of both the *ΔVdTHI20-2* and *ΔVdTHI20-6* strains was delayed compared with that of *Vd-wt* and both complemented strains (Figure 3A). In addition, the colony diameters of the *ΔVdTHI20* mutants on different carbon sources were smaller than those of *Vd-wt* and the two complemented strains *ΔVdTHI20-2-c* and *ΔVdTHI20-6-c*. The colony diameters of the *ΔVdTHI20* mutants on different carbon sources, sucrose, pectin, starch, galactose, and xylose, were reduced to 68.7%, 63.23%, 69.69%, 54.76%, and 59.76%, respectively, of the diameter of *Vd-wt* (Figure 3B–F). Similarly, in the *ΔVdThit* mutant, a disruptant of *VdThit*, radial mycelial growth was also significantly reduced [20]. These results indicate that *VdTHI20* contributes to the hyphal vegetative growth and carbon utilization of *V. dahliae*.

### 2.3. Exogenous Thiamine Recovered the Hyphal Growth of ΔVdTHI20 Mutants

Hyphal growth in the *ΔVdTHI20* mutant strains was slower than that in the *Vd-wt* strain and complemented strains (*ΔVdTHI20-2* and *ΔVdTHI20-6*) according to our previous study. To uncover whether exogenous thiamine could recover the hyphal growth of the *ΔVdTHI20* mutants, MM agar plates were supplied with different concentrations of exogenous thiamine (0.2 mg/L to 2 mg/L), and the colony diameters were measured at 4 and 8 days. Hyphal growth was recovered with a high concentration of exogenous thiamine (Figure 4A). This result is similar to previous studies on *ΔVdTHI4(VdTHI4* involved in HET-P synthesis). We then confirmed the expression level of the *VdThit* gene (encoding thiamine transport protein) in the *Vd-wt* strain and *∆VdTHI20* mutant strains with exogenous thiamine. The expression level of *VdThit* was significantly higher in the *∆VdTHI20* mutant strains than in *Vd-wt*. Interestingly, the expression level of *VdThit* in the *Vd-wt* strain and *∆VdTHI20* mutant strains showed no significant difference with different concentrations of exogenous thiamine (Figure 4B). These results suggest that thiamine can enter *V. dahliae* strains through the plasma membrane by simple diffusion and that *V. dahliae* strains can use thiamine in two separate pathways, each of which is indispensable.

### 2.4. Knockout of VdTHI20 Reduced Conidial Germination and Production and Caused Abnormal V. dahliae Hyphal Morphology

When conidia and hyphae were observed with the optical microscope after being cultured for 24 h after germination in liquid Complete medium (CM) broth, *Vd-wt* hyphae displayed radical growth; however, *ΔVdTHI20* hyphae grew slowly and generated swollen and atypical branches (Figure 5A). In addition, the role of *ΔVdTHI20* in conidial germination was analysed. *Vd-wt*, the two mutants (*ΔVdTHI20-2* and *ΔVdTHI20-6*) and the two complemented stains (*ΔVdTHI20-2-c* and *ΔVdTHI20-6-c*) were incubated in liquid CM for 30 h. The conidial germination of the *ΔVdTHI20* mutants occurred more slowly than that of *Vd-wt* and both complemented stains, *ΔVdTHI20-2-c* and *ΔVdTHI20-6-c*. Only approximately 36% of the conidia of the *ΔVdTHI20* mutant strains germinated, while 90% of *Vd-wt* conidia and 85% of the complemented strains conidia germinated (Figure 5B). Additionally, conidia production was studied in all *V. dahliae* strains cultured on Czapek–Dox agar plates. *ΔVdTHI20-2* and *ΔVdTHI20-6* produced significantly fewer conidia than *Vd-wt* and the complemented strains (Figure 5C). The results suggest that *VdTHI20* participates in conidial germination, conidiation, and mycelial development.

### 2.5. Loss of VdTHI20 Resulted in Reduced Tolerance to UV Damage

Thiamine was reported to play a significant role in oxidative stress response [22,23,24,25,26]. The orthologue of HET-P biosynthesis gene *THI4*, *sti35* was reported to play a role in the oxidative stress response in *Fusarium oxysporum* [27]. *VdTHI4* was reported to affect UV damage repair in *S. cerevisiae* and *V. dahlia* [19]. In addition, *Vd*∆*THI4* mutants were susceptible to oxidative stressors menadione and 2,4-DAPG. The *VdΔThit* mutants were also susceptible to UV damage and oxidative stressors menadione [20].

The function of *VdTHI20* in response to UV radiation stress was evaluated. The number of surviving *ΔVdTHI20* mutant colonies after exposure to UV irradiation was significantly reduced compared with the number of surviving *Vd-wt* and complemented *strain* colonies (*P* < 0.01). This result is similar to the previous studies on *ΔVdTHI4* and *VdΔThit* [19]. Specifically, only 1.2% of the UV-treated *ΔVdTHI20* mutant colonies survived UV irradiation at 50 J/m^2^, compared with 11.5% of *Vd-wt* colonies (Figure 6).

### 2.6. VdTHI20 Is Crucial for the Pathogenicity of V. dahliae towards Plants

We then evaluated the *VdTHI20* deletion effect on virulence. Wild-type *N. benthamiana* seedlings were inoculated with a conidial suspension of *Vd-wt*, *ΔVdTHI20* mutant strains (*ΔVdTHI20-2* and *ΔVdTHI20-6*), and complemented strains (*ΔVdTHI20-2-c* and *ΔVdTHI20-6-c*) by dipping intact roots. At 15 days post-inoculation (dpi), the seedlings infected with the *ΔVdTHI20* mutant strains presented moderate symptoms, with only mild interveinal chlorosis of leaves and nearly no necrosis. In contrast, the seedlings infected with *Vd-wt* and the complemented strains displayed typical symptoms, including stunting, chlorosis and wilting, and appeared to be nearly dead (Figure 7A,B). The disease level of *N. benthamiana* was rescued significantly by *VdTHI20* deletion at 5 dpi, 10 dpi and 15 dpi (Figure 7C). These results are similar to the previous studies on *VdΔThit* [19]. In addition, the concentration of fungal DNA amplified from *N. benthamiana* seedlings inoculated with the *ΔVdTHI20* mutant strains was lower than that amplified from the *Vd-wt* and complemented strains (Figure 7D).

### 2.7. Fungal Colonization and Root Infection with the ΔVdTHI20 Mutant Were Impaired

To investigate fungal colonization and root infection of the *ΔVdTHI20* mutant strains, the GFP-labelled strains *VdΔTHI20-2-GFP* and *Vd-wt-GFP* constitutively expressing eGFP were used. Thus, fungal colonization and penetration were observed under LSCM. Hyphae penetrated the epidermal and vascular system cells with the *Vd-wt-GFP* strain at 3 dpi. Fewer hyphae of the *ΔVdTHI20-2-GFP* mutant strain germinated and colonized *N. benthamiana* roots than the *Vd-wt-GFP* strain (Figure 8). This result is similar to the previous studies on *VdΔThit* [19]. These observations demonstrated that *VdTHI20* contributes to fungal penetration and colonization.

### 2.8. DsRNA of VdTHI20 Confers Resistance Against V. dahliae in Transgenic N. benthamiana Lines

To validate whether dsVdTHI20 plays a role in resistance against *V. dahliae*, wild-type and transgenic *N. benthamiana* lines (RNAi-THI20-3, RNAi-THI20-8 and RNAi-THI20-12) were inoculated with a conidial suspension of *Vd-wt*. At 15 dpi, the seedlings of wild-type *N. benthamiana* displayed typical symptoms and were nearly dead. In contrast, the seedlings of transgenic *N. benthamiana* lines showed weak symptoms (Figure 9A). The disease level of transgenic *N. benthamiana* lines was significantly lower than that of wild-type *N. benthamiana* at 5 dpi, 10 dpi, and 15 dpi (Figure 9B). In addition, the concentration of fungal DNA in transgenic *N. benthamiana* lines infected with the conidial suspension of *Vd-wt* was significantly lower than that in wild-type *N. benthamiana* (Figure 9C). These results indicate that VdTHI20 is required for the pathogenicity of *V. dahliae*.

## 3. Discussion

Severe losses in cotton production are caused by verticillium wilt every year. Until now, the pathogenesis mechanism has been unclear, making disease control challenging. Verticillium wilt of cotton is caused by *V. dahliae*, a fungal plant pathogen. Thiamine was demonstrated to contribute to fungal pathogenicity based on a previous study [26]. The THI20 family proteins were reported to be involved in the formation of the pyrimidine moiety of thiamine [28]. In our study, we characterized the functions of a member of the THI20 family, VdTHI20, in the pathogenicity of *V. dahliae*. The deletion of *VdTHI20* caused a reduction in mycelial growth, conidial germination and production, and UV resistance. The invasion and colonization abilities of *ΔVdTHI20* mutants also decreased. The *ΔVdTHI20* mutant strains exhibited reduced pathogenicity. Together, our research supports that VdTHI20 is necessary for infection by *V. dahliae*.

Previous studies indicated that VdTHI4 is required for the pathogenicity of *V. dahliae* in tomato. VdTHI4, which is involved in thiazole (a part of thiamine) biosynthesis [29], plays a key role in the conidial growth, infection and colonization of *V. dahliae.* Supplemented thiamine can restore the growth of the *VdΔTHI4* deletion strain on thiamine-free medium [19]. Another team found that the plasma membrane protein VdThit is also essential for the pathogenicity of *V. dahliae*. This thiamine transporter protein, VdThit, has an important influence on the vegetative growth, conidial germination and production, initial infection and root colonization of *V. dahliae*. By supplementing exogenous thiamine, the defects of *VdΔThit* mutants in growth, conidiation, and virulence were restored [20]. Similarly, exogenous thiamine restored the growth of *ΔVdTHI20* mutant strains. According to the results of this study and previous studies, *V. dahliae* can use self-synthesized thiamine or uptake thiamine from the environment. These two approaches for thiamine utilization are irreplaceable for the virulence of *V. dahliae*.

The thiamine biosynthesis pathway *THI4* gene orthologue *sti35* was reported to play a role in the oxidative stress response in *Fusarium oxysporum* [27]. THI4 affected UV damage repair in *S. cerevisiae* and *V. dahliae*. Additionally, *Vd*∆*THI4* exhibited reduced growth on medium with the oxidative stressors menadione and 2,4-DAPG [19]. Fewer colonies of *VdΔThit* mutants than of wild type survived after treatment with UV radiation, and *VdΔThit* mutant strains grew slower than wild type when supplemented with menadione [20]. In accordance with this research, our results showed that *ΔVdTHI20* mutants were more susceptible to UV stress.

Our results indicate that *ΔVdTHI20* has a significant contribution to UV radiation tolerance. Our findings are consistent with previous studies on *ΔVdTHI4* (involved in thiamine synthesis pathway) and *ΔVdThit* (a thiamine transport protein) mutant strains that could not repair damage from UV radiation [19,20].

RNA interference (RNAi) is a technology using dsRNA to inhibit the expression of target genes [30,31,32]. In recent years, RNAi has been proven to be a powerful tool to improve the resistance of plants [33,34,35]. Host-induced gene silencing (HIGS) technology, trans-kingdom RNA silencing, has been developed and applied to protect crops against fungal, pest and viroid infections. Transgenic corn plants harbouring western corn rootworm (WCR) dsRNA were able to control coleopteran pests [36]. RNAi-mediated silencing of *OsSSI2* enhanced the resistance to blast and leaf blight in rice [37]. Tomato plants transformed with a viroid hairpin RNA construct showed resistance to potato spindle tuber viroid infection [38]. Expressing dsRNA in tobacco plants improved pest resistance [39]. The silencing of the *Foc* TR4 *ERG6/11* genes increased the resistance of banana to Fusarium wilt [40]. In eggplant, HIGS of Mi-msp-1 improved nematode resistance [41].

*V. dahliae* infects over 200 plant species in nature and has become the most notorious plant vascular pathogen [5,42]. Due to its long-term survival in soil and tendency for host vascular colonization, *V. dahliae* is very difficult to control with fungicides, and few plant resistance genes have been identified thus far. For cash crops, such as cotton and sunflower, and major food crops, such as potato, no resistance gene has been identified that efficiently protects plants against *V. dahliae* infection [43]. Based on this fact, HIGS technology has shown promising potential for application against this soilborne fungus [44,45,46,47,48].

Since HIGS has been successfully applied to protect plants against fungal pathogens [49,50,51,52,53], it is not surprising that plants were able to export exogenous artificial sRNAs to target fungal gene transcripts. Subsequently, sRNAs isolated from *V. dahliae* recovered from infected cotton plants were sequenced, and in total, 28 different cotton sRNA sequences were identified. This proved that host-derived sRNAs were indeed transmitted into the fungal pathogen during infection. It was also the first experimental evidence of natural transmission of trans-kingdom RNAi signals from host to fungal pathogen and the cleavage of fungal virulence genes [53]. Researchers have proven that siRNAs and dsRNAs in extracellular vesicles can be efficiently transferred into fungal cells to depress invasion by feeding on plants and parasitism [54].

In this study, transgenic *N. benthamiana* lines expressing dsVdTHI20 exhibited milder disease symptoms and had lower disease levels and less fungal DNA than wild-type *N. benthamiana*. The pathogenesis-related gene *VdTHI20* exhibits the potential to control *V. dahliae* for important crops and provides new insight and a valuable idea for the development of comprehensive management strategies for *V. dahliae*.

## 4. Materials and Methods

### 4.1. Fungal Strains, Plant Material and Culture Conditions

The wild-type *V. dahliae* strain 991 (*Vd-wt*) is preserved in our laboratory. Complete medium (CM, 6 g/L yeast extract, 6 g/L casein acid hydrolysate, 10 g/L sucrose) and Czapek–Dox (3.0 g/L NaNO_3_, 0.5 g/L MgSO_4_·7H_2_O, 0.5 g/L KCl, 0.01 g/L FeSO_4_·7H_2_O, 1.0 g/L K_2_HPO_4_, 15 g/L agar) agar were used in this study for the *Vd-wt*, *ΔVdTHI20* mutant and complemented strains. *N. benthamiana* seedlings were planted in a growth chamber at 23–25 °C with a photoperiod of 16 h day/8 h night and a relative humidity of 60%–70%. *Agrobacterium tumefaciens* AGL1 and the plasmids pGKO2, pCAM-neo, pCM-Hyg, and pCAMgfp were obtained from Prof. Xiaofeng Dai from the Institute of Agro-products Processing Science and Technology, Chinese Academy of Agricultural Sciences in Beijing, China.

### 4.2. Plasmid Construction and Fungal Transformation

For *VdTHI20* knockout cassette construction, In-Fusion cloning was used as previously described [55]. The neomycin phosphotransferase resistance (*neo*) cassette was amplified from pCAM-neo using the primer pair neo-F and neo-R (Table 2). The 1197 bp 5′ upstream sequence and 1108 bp 3′ downstream sequence of *VdTHI20* were amplified from the *Vd-wt* strain genome with the primer pairs VdTHI20-5F and VdTHI20-5R and VdTHI20-3F and VdTHI20-3R, respectively (Table 2). For In-Fusion cloning, the vector pGKO2 was digested with *Eco*RI or *Hin*dIII to create linearized pGKO2 (Figure 2A). According to the specifications of the In-Fusion dry-down PCR cloning kit (Clontech, Mountain View, CA, USA), the linearized vector together with three amplicons, the *neo* cassette, the 5′ upstream fragment and the 3′ downstream fragment, were fused to generate the pGn-VdTHI20 plasmid pGKO2 [*Eco*RI]::VdTHI20-5′::neo::VdTHI20-3′::pGKO2 [*Hin*dIII]. All the constructs were verified by enzymatic identification and DNA sequencing (Sangon, Shanghai, China).

The *VdTHI20* complementary vector pCM-Hyg-VdTHI20 was generated using the same method. The pCM-Hyg vector with the hygromycin B resistance gene (*hph*) cassette was digested with the enzymes *Kpn*I and *Xba*I. Three fragments, full-length *VdTHI20* cDNA, the TrpC promoter and the Nos terminator, were individually amplified with the primer pairs C-VdTHI20-F and C-VdTHI20-R, C-TrpC-F and C-TrpC-R and C-Nos-F and C-Nos-R (Table 2). All three amplicons were inserted into a digested pCM-Hyg vector with In-Fusion enzyme.

The constructs pGn-VdTHI20 were introduced into *A. tumefaciens* AGL1 using electroporation. *Agrobacterium tumefaciens*-mediated transformation (ATMT) was used as described previously [56]. The homologous recombination transformants were selected on PDA medium (potato, 200 g/L, glucose, 20 g/L, agar, 15 g/L) with 200 μg/mL of cefotaxime, 50 μg/mL of G418 and 200 μg/mL of 5-fluoro-2′-deoxyuridine. The complemented transformants were screened on the PDA medium with 200 μg/mL of cefotaxime and 50 μg/mL of hygromycin, in addition to 100 μg/mL geneticin for complementary transformants of *VdTHI20* deletion mutants. Likewise, the plasmid pCAMgfp containing the enhanced green fluorescent protein (eGFP) and *hph* genes was introduced into the *Vd-wt* strain and the *ΔVdTHI20* mutant strains using the same method. The single spore isolation was performed for all transformants, and the positive transformants were verified by PCR with the test primers (Table 2).

### 4.3. Confirmation of VdTHI20 Gene Disruption or the Complementation of ΔVdTHI20 and Screening for GFP-tagged Strains

The primer pairs VdTHI20-J-F and VdTHI20-J-R were used to examine the *neo* cassette of the *VdTHI20* knockout mutants (Table 2). Complemented *strains* were checked by PCR using the primer pairs VdTHI20-J-F and VdTHI20-J-R and hyg-F and hyg-R (Table 2). Additionally, GFP-labelled strains were screened for the *hph* gene using the primer pair hyg-F and hyg-R. GFP fluorescence was observed under a laser confocal scanning microscope (LSCM; LSM700, Zeiss, Jena, Germany).

### 4.4. Growth, Conidia Production, and Germination Assays

The colony diameter and morphology of the *Vd-wt* strain, *ΔVdTHI20* mutant strains, and complemented *ΔVdTHI20* strains were recorded every 2 to 3 days on minimal medium (MM) agar (6 g/L NaNO_3_, 0.52 g/L KCl, 0.152 g/L MgSO_4_, 1.52 g/L KH_2_PO_4_, 0.01 g/L thiamine, 10 mL/L trace elements, and 15 g/L agar) amended with different carbon sources (10 g/L pectin, 10 g/L galactose, 10 g/L xylose, 10 g/L starch, or 30 g/L sucrose) [57].

For conidia production tests, conidial suspension (1 mL, 5 × 10^6^ conidia/mL) of *Vd-wt* strain, *ΔVdTHI20* mutant strains and complemented strains were added to 200 mL Czapek–Dox broth in sterile conical flasks. After 6 days on a shaker with 180 rpm at 25 °C, the suspension was filtered through a cell strainer, and then the conidia were counted using a hemacytometer and a light microscope. Conidia production was then calculated as previously described [58]. Each assay was repeated three times.

For germination tests, the germination of all *V. dahliae* strains was calculated as described previously [59]. The conidial suspension of *Vd-wt* strain, *ΔVdTHI20* mutant strains and complemented *∆VdTHI20* mutant strains were cultured in CM broth, and then shaken for 12 h with 140 rpm at 25 °C. Samples were examined at 2 h intervals and germination rate was determined by counting 100 conidia. Each assay was repeated three times.

### 4.5. Phenotypic Analysis under Stress Conditions

Conidial suspensions of each strain, the *Vd-wt* strain, *ΔVdTHI20* mutant strains, and complemented strains, were spread onto Czapek–Dox agar and incubated at 25 °C for 1 h. For UV light treatment, conidia from each strain were placed under UV radiation at 50, 100, 150, and 200 J/m^2^ with a UV Crosslinker Spectrolinker XL-1000A (Spectronics Corp., Westbury, NY, U.S.A.), after which the conidia were incubated at 25 °C for three days. Colonies were enumerated on each plate. The experiment was repeated three times.

### 4.6. Quantitative Real-Time PCR (qRT-PCR)

The expression level of the gene *VdThit* (VDAG_03620) encoding a thiamine transport protein was measured in the *Vd-wt* strain, *ΔVdTHI20* mutant strains, and complemented *ΔVdTHI20-c* strains. Total RNA was extracted from each *V. dahliae* strain using the RNA miniprep kit (Axygen, Union City, CA, USA) according to the manufacturer’s instructions. cDNAs were synthesized using a Toyobo RT kit (Osaka, Japan). qRT-PCR was performed with the SYBR Fast qPCR kit (Kapa Biosystems, Boston, MA, USA). The expression level of *VdThit* was quantified using the β-tubulin gene (DQ266153) as an internal reference, which was amplified with the primer pair VdThit-F and VdThit-R (Table 2). The qRT-PCRs were performed on an ABI QuantStudio 6 flex PCR thermocycler (Applied Biosystems, Foster City, CA, USA) and repeated three times.

### 4.7. Hyphal Growth under Different Concentrations of Exogenous Thiamine

The colony diameter, conidia production, and virulence of the *ΔVdTHI20* mutant strains were characterized as described above. No thiamine or 0.2 to 2 mg/L exogenous thiamine was added to each Czapek–Dox agar plate. The assay was repeated three times independently.

### 4.8. Microscopic Observation of Initial Infection

For microscopically observing initial infection, roots of *N. benthamiana* seedlings with 6–7 true leaves were infected with the *Vd-wt-GFP* strain and the *VdΔTHI20-2-GFP* mutant strain. After 3 days, the infected slices of *N. benthamiana* roots were observed under LSCM. The assay was repeated three times independently.

### 4.9. Plasmid Construction and Plant Transformation

The primer pair was designed based on the *THI20* ORF (Trans-THI20, Table 2). The partial BP adaptors were used for targeted fragment amplification. The whole sequence was amplified with BP site primers, followed by inserting into the pDONR207 vector by a BP recombination reaction (Invitrogen, Carlsbad, CA, USA). The targeted fragment was then inserted into the pK7GWIWG2(I) vector to obtain pK7GW12WG2-THI20 by an LR recombination reaction (Invitrogen, Carlsbad, CA, USA). The resulting plasmid was confirmed by sequencing, followed by transforming into Agrobacterium tumefaciens strain LBA4404 by electroporation.

Seeds of *N*. *benthamiana* were grown on MS agar. After 30 days, sterile leaves were immersed in a solution of *A*. *tumefaciens* strain LBA4404 with the recombinant plasmid and cultured on MS agar plates subsequently. After 3 days, the leaves were transferred to MS agar plates with 100 mg/L kanamycin.

## Figures and Tables

**Figure 1 ijms-21-01378-f001:**
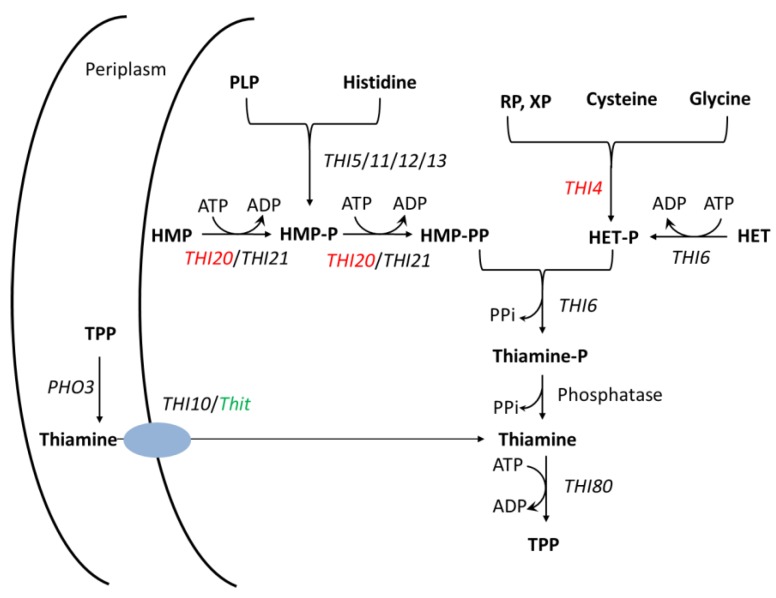
A schematic of the thiamine biosynthesis pathway. Black italics represent the genes found in *S. cerevisiae*. Green italics represent the genes found in *V. dahliae*. Red italics indicate genes found in both *S. cerevisiae* and *V. dahliae*. Blue oval represents thiamine transport protein.

**Figure 2 ijms-21-01378-f002:**
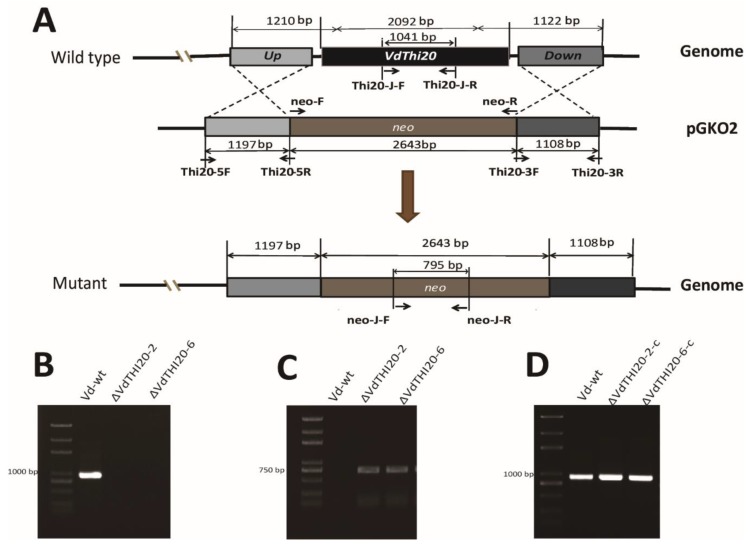
Schematic of the *VdTHI20* gene knock out vector design and complemented *ΔVdTHI20* mutant strains. (**A**) Construction scheme designed for the *VdTHI20* gene knockout vector. *VdTHI20* gene deletion was verified by PCR. Genomic DNA of the wild-type *V. dahliae* strain 991 (*Vd-wt*) and *ΔVdTHI20* mutant strains were amplified with the primer pairs neo-J-F and neo-J-R and THI20-J-F and THI20-J-R. (**B**) A 1041 bp fragment was amplified from *Vd-wt* using the primer pair THI20-J-F and THI20-J-R. (**C**) A 795 bp fragment was amplified with the primer pair neo-J-F and neo-J-R from the *VdΔTHI20* mutant strains. (**D**) Confirmation of complemented stains (*ΔVdTHI20-2-c* and *ΔVdTHI20-6-c*) was performed with the primers THI20-J-F and THI20-J-R.

**Figure 3 ijms-21-01378-f003:**
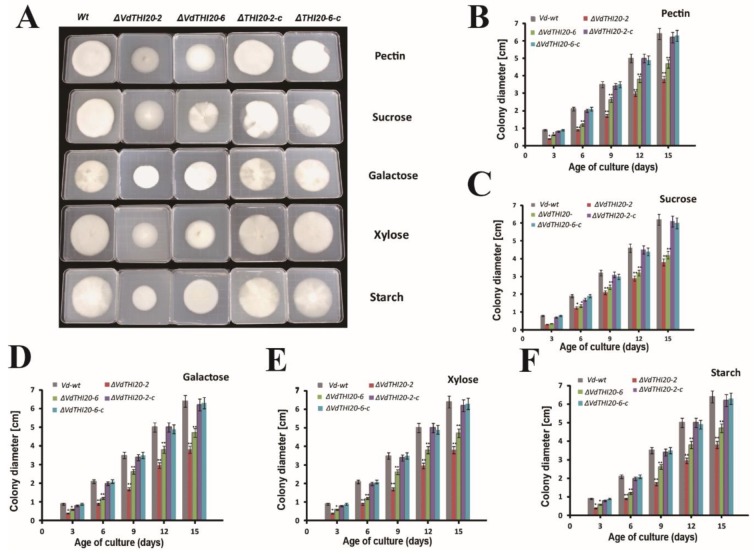
Mycelial growth of the *Vd-wt* strain, *ΔVdTHI20* (*ΔVdTHI20-2* and *ΔVdTHI20-6*) mutant strains, and complemented strains (*ΔVdTHI20-2-c* and *ΔVdTHI20-6-c*). (**A**) Colony morphology was observed on MM agar with different carbon sources after 15 days. (**B**–**F**) Colony diameters of *Vd-wt* strain, *ΔVdTHI20* mutant strains and complemented strains grown on different carbon sources, sucrose, pectin, starch, galactose, and xylose, respectively. Colony diameters were measured after 3, 6, 9, 12, and 15 days. Values represent the mean ± standard deviation from three independent replicates. Mycelial growth of the *ΔVdTHI20* mutant strains and the complemented strains on MM agar were compared with *Vd-wt* strain. Asterisks (*) indicate significant differences by Student’s *t* test at *P* < 0.05. Asterisks (**) indicate significant differences by Student’s *t* test at *P* < 0.01.

**Figure 4 ijms-21-01378-f004:**
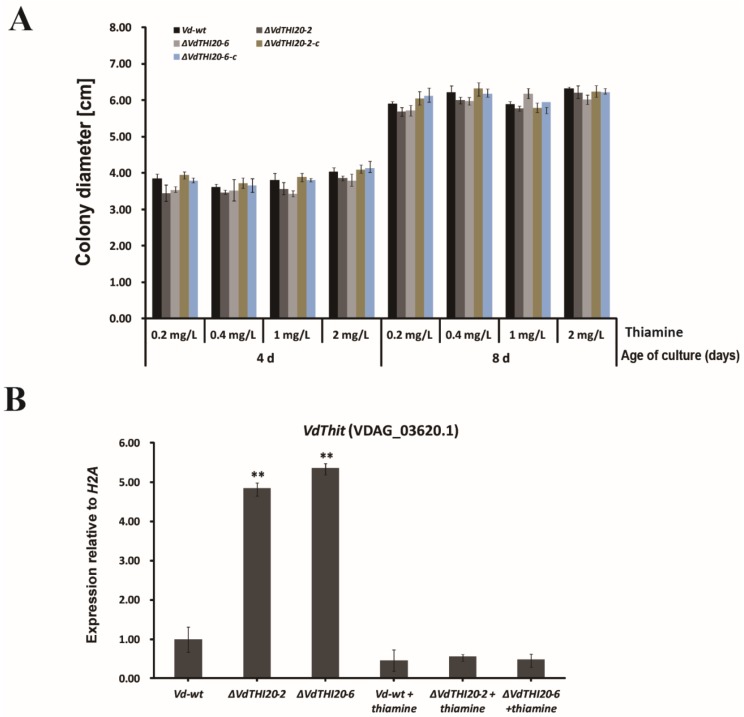
Exogenous thiamine recovered the hyphal growth of *ΔVdTHI20* mutants on Czapek–Dox agar plates. (**A**) Colony diameters of the *Vd-wt* strain, *ΔVdTHI20* mutant strains (*ΔVdTHI20-2* and *ΔVdTHI20-6*), and complemented strains (*ΔVdTHI20-2-c* and *ΔVdTHI20-6-c*) supplemented with exogenous thiamine (0.2 mg/L to 2 mg/L) were measured at 7 dpi. (**B**) The expression of *VdThit* in the *Vd-wt* strain and *ΔVdTHI20* mutant strains with or without exogenous thiamine. Values are means ± standard deviation of three independent experiments performed in duplicate. Asterisks (**) indicate significant differences by Student’s *t* test at *P* < 0.01.

**Figure 5 ijms-21-01378-f005:**
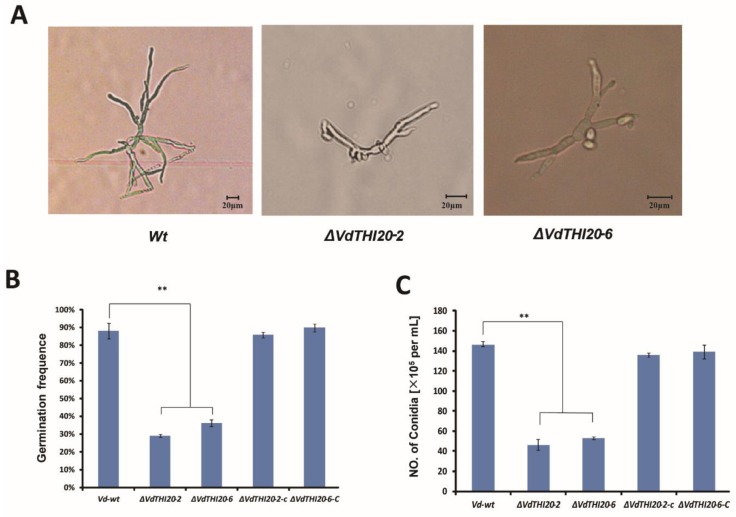
Deletion of *VdTHI20* influences hyphal morphology, conidial germination and production. (**A**) Hyphae of each strain 24 h after germination in liquid CM broth. The hyphae of the *ΔVdTHI20* strains branched irregularly and did not form whorled branches. The hyphae of the *ΔVdTHI20* strains were also more swollen than those of *Vd-wt*. (**B**) Percentages of germinated conidia of *Vd-wt*, *ΔVdTHI20* mutant strains (*ΔVdTHI20-2* and *ΔVdTHI20-6*), and complemented strains (*ΔVdTHI20-2-c* and *ΔVdTHI20-6-c*) 30 h after inoculation in liquid CM. Reduced germination frequency of the *ΔVdTHI20* mutant strains in liquid CM in comparison with the *Vd-wt* strain is significant while the complemented *∆VdTHI20* mutant strains in liquid CM in comparison with the *Vd-wt* strain is not significant at 30 h. (**C**) Number of conidia produced by the *Vd-wt* strain, *ΔVdTHI20* mutant strains (*ΔVdTHI20-2* and *ΔVdTHI20-6*), and complemented strains (*ΔVdTHI20-2-c* and *ΔVdTHI20-6-c*) after 15 days on Czapek–Dox agar plates. Decreased number of conidia produced by the *ΔVdTHI20* mutant strains the complemented strains on Czapek–Dox agar plates in comparison to the *Vd-wt* strain is significant and not significant after 15 days, respectively. Values represent the mean ± standard deviation from three independent replicates. Asterisks (**) indicate significant differences by Student’s *t* test at *P* < 0.01.

**Figure 6 ijms-21-01378-f006:**
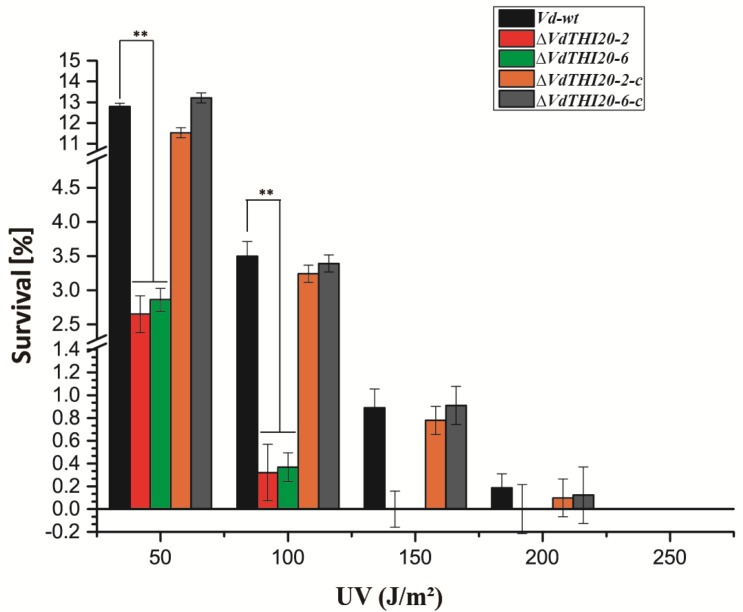
Effect of UV radiation on the growth of the *Vd-wt* strain, *ΔVdTHI20* mutant strains (*ΔVdTHI20-2* and *ΔVdTHI20-6*), and complemented strains (*ΔVdTHI20-2-c* and *ΔVdTHI20-6-c*). The *ΔVdTHI20* mutant strains were hypersensitive to UV radiation compared with the *Vd-wt* and complemented strains. The percentages of surviving colonies were computed after UV exposure at 50, 100, 150, and 200 J/m^2^. The reduction of survival of the *ΔVdTHI20* mutant strain colonies after UV-treatment at 50 and 100 J/m^2^ in comparison to that of the *Vd-wt* strain is significant. The reduction is restored in the complemented *∆VdTHI20* mutant strains. Statistical analyses of survival indicated no significant difference between the complemented strains and the *Vd-wt* strain. Values represent the mean ± standard deviation from three independent replicates. Asterisks (**) indicate significant differences by Student’s *t* test at *P* < 0.01.

**Figure 7 ijms-21-01378-f007:**
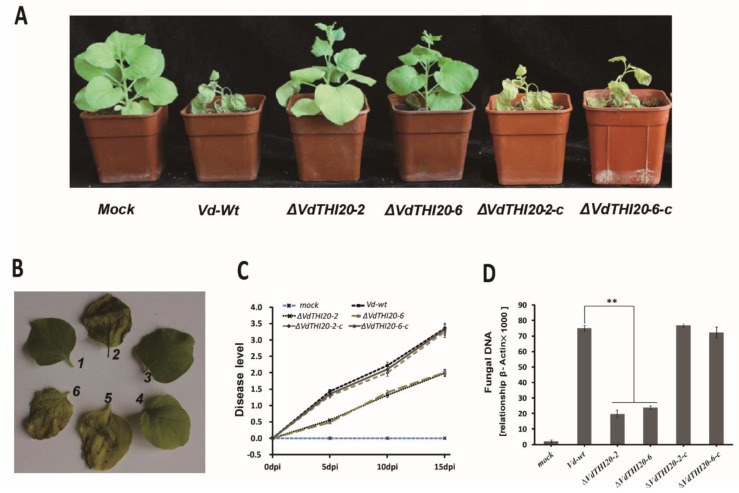
Virulence analysis of the *Vd-wt* strain, *ΔVdTHI20* mutant strains (*ΔVdTHI20-2* and *ΔVdTHI20-6*), and complemented *∆VdTHI20* mutant strains (*ΔVdTHI20-2-c* and *ΔVdTHI20-6-c*) on *N. benthamiana*. (**A**) Virulence phenotypes of *N. benthamiana* seedlings at 12 dpi with different *V. dahliae* strains. (**B**) Representative leaf images of *N. benthamiana* at 12 dpi with different *V. dahliae* strains. 1, mock; 2, representative leaf of *N. benthamiana* infected with the *Vd-wt* strain; 3,4, representative leaf of *N. benthamiana* infected with *ΔVdTHI20-2* and *ΔVdTHI20-6* respectively; 5,6, representative leaf of *N. benthamiana* infected with *ΔVdTHI20-2-c* and *ΔVdTHI20-6-c* respectively. (**C**) Disease level of *N. benthamiana* seedlings after a 2-min root dpi in a conidial suspension of the different *V. dahliae* strains at 5, 10, and 15 dpi. (**D**) Fungal DNA concentration in *N. benthamiana* hypocotyls with different *V. dahliae* strains at 12 dpi. *N. benthamiana* seedlings inoculated with tap water (mock) as a control. The difference of fungal DNA concentration between the *ΔVdTHI20* mutant strains and the *Vd-wt* strain is significant. The difference of fungal DNA concentration between the complemented *∆VdTHI20* mutant strains and the *Vd-wt* strain is not significant. Values represent the mean ± standard deviation from three independent replicates. Asterisks (**) indicate significant differences by Student’s *t* test at *P* < 0.01.

**Figure 8 ijms-21-01378-f008:**
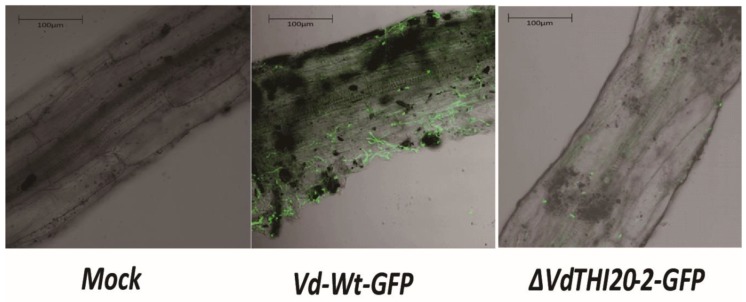
Green fluorescent protein (GFP) fluorescence detection of fungal colonization and root infection in *N. benthamiana* with the *Vd-wt-GFP* strain, the *ΔVdTHI20-2-GFP* mutant strains, and sterile water (mock) at 3 dpi.

**Figure 9 ijms-21-01378-f009:**
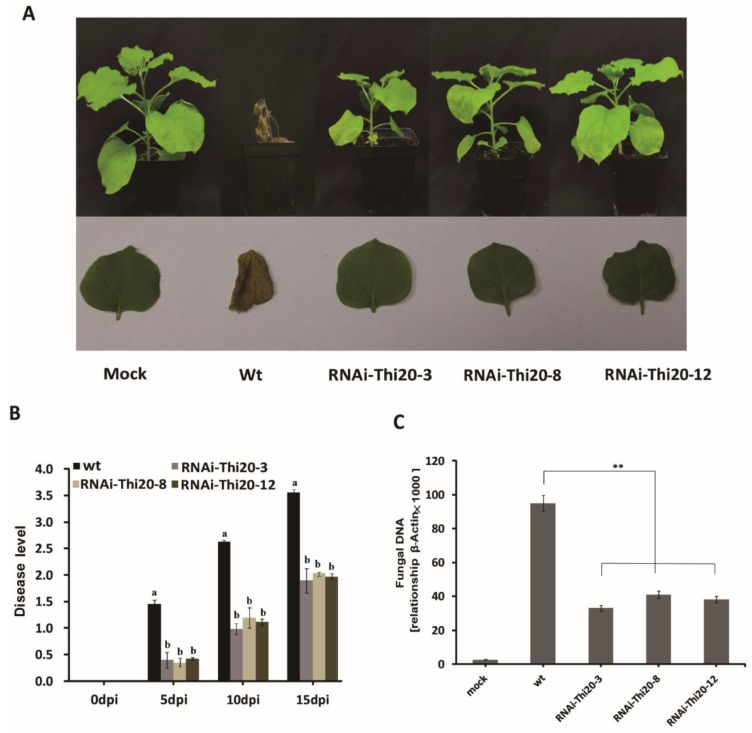
Virulence analysis of the *Vd-wt* strain in wild-type and transgenic *N. benthamiana* lines (RNAi-THI20-3, RNAi-THI20-8, and RNAi-THI20-12). (**A**) Virulence phenotypes of wild-type and transgenic *N. benthamiana* lines and representative leaves. (**B**) Disease level of wild-type and transgenic *N. benthamiana* lines after a 2-min root dpi in conidia of *Vd-wt* at 5, 10, and 15 dpi. (**C**) Fungal DNA concentration in wild-type and transgenic *N. benthamiana* hypocotyls infected with *Vd-wt* at 12 dpi. *N. benthamiana* seedlings inoculated with tap water (mock) as a control. Values represent the mean ± standard deviation from three independent replicates. Asterisks (**) indicate significant differences by Student’s *t* test at *P* < 0.01.

**Table 1 ijms-21-01378-t001:** Genes involved in thiamine biosynthesis.

Gene	Protein Function	References
*THI4*	HET-P synthesis; Stress response and DNA repair	[14,19]
*THI5/THI11/THI12/THI13*	HMP-P synthesis	[10]
*THI6*	Thiamin-phosphate pyrophosphorylase and hydroxyethylthiazole kinase	[15]
*THI10*	Thiamine transport protein	[17]
*THI20*	HMP-PP synthesis and thiamine degradation	[11,12,21]
*THI21/THI22*	HMP-PP synthesis	[11,12]
*THI80*	Thiamin pyrophosphokinase	[16]
*VdThit*	Thiamine transport protein; Vegetable growth, reproduction and invasive hyphal growth	[20]
*PHO3*	Thiamin phosphates hydrolysis	[18]

**Table 2 ijms-21-01378-t002:** Primers used in this study.

Primer Name Sequences (5′→3′)	
Thi20-5F	GTACCCAATTCGAATTC ATTGTGTTTGAGGAGGACACCGAT
Thi20-5R	CAAGACAGCCCGCAAAC TCTCCTTGAGAAAACGAGTGA
Thi20-3F	CCCAGAATGCACAGGT TGTTGATGTCGGTGTCATCGTC
Thi20-3R	GACGGTATCGATAAGCTT TGTTGGAAAAAGGTCAGTCAT
C-TrpC-F	TTGAAGGAGCATTTTTGGGC
C-TrpC-R	ATCGATGCTTGGGTAGAATAGGT
C-VdThi20-F	AAAAGTACTATGGCACAGCAGATGGGCCG
C-VdThi20-R	AAACTGCAGTCTACTGGCACGGGAACATCT
C-Nos-F	AGATGCCGACCGGGATCCACTT
C-Nos-R	TTATCTTTGCGAACCCAGGG
neo-F	GTTTGCGGGCTGTCTTGACG
neo-R	TACCTGTGCATTCTGGGTAA
Thi20-J-F	GCGCAGGACACAAAGGGCGT
Thi20-J-R	AGCGTGCCGTTGCCGAGACC
noe-J-F	ATGATTGAACAAGATGGATT
noe-J-R	TCAGAAGAACTCGTCAAGAA
Thit-F	CTCGTGACTTTATCGGGTTTCT
Thit-R	GGCGGATGAGCTGGAATTAT
VdBt-F	TTCCCCCGTCTCCACTTCTTCATG
VdBt-R	GACGAGATCGT TCATGTTGAACTC
Nb-actin-F	GGACCTTTATGGAAACATTGTGCTCAGT
Nb-actin-R	CCAAGATAGAACCTCCAATCCAGACAC
Vd-F	CCGCCGGTCCATCAGTCTCTCTGTTTATAC
Vd-R	CGCCTGCGGGACTCCGATGCGAGCTGTAAC
TransThi20F	GGGGACAAGTTTGTACAAAAAAGCAGGCTCGCAACCATTGTCAAGCACA
TransThi20R	GGGGACCACTTTGTACAAGAAAGCTGGGTTTCCAGTACACGTTGCCCTC

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
