# Peer review of "Verticillium dahliae VdTHI20, Involved in Pyrimidine Biosynthesis, Is Required for DNA Repair Functions and Pathogenicity"

_ijms, 2020, doi:10.3390/ijms21041378_

Round 1
Reviewer 1 Report
The authors present the deletion of a gene involved in the thiamine biosynthesis pathway in Verticillium dahliae. They show that deletion of the gene VdTHI20 causes impaired vegetative growth, conidia production and germination, impaired root colonization and virulence. Especially the HIGS experiment is promising. The work is overall well done. The method section lacks some details that must be added.
Major:
Please revise the list of authors: I don´t think it should be general policy that revision of a manuscript qualifies for authorship. The same applies for a participation in material collection. In the presented work material collection does not play a major (not even a minor) role, if I am not mistaken.
110/135/149/174 and other statistics, please indicate in the legend and the graphs which two samples were compared in each students t test. (wild type vs each mutant and complemented strain)
Figure 3. Scale bars are missing.
Please connect the observed contribution to UV radiation tolerance to thiamine biosynthesis.
Conidia formation, germination defects and cell morphology defects often are found in connection with cell wall integrity mutants. Please check sensitivity of the VdTHI20 mutants and complemented strains in comparison to wild type for sensitivity to cell-wall- perturbing agents like CFW, Congo Red to get an idea if the reduced virulence might be connected to disruption in cell wall integrity, which in turn might trigger plant defence.
Materials and Methods:
314: Please comment on the verification of the vector- was the vector sequenced or confirmed by restriction mapping?
321 ff. Give details on ATMT transformation procedure
328: Gene disruption and replacement are different methods, with replacement being the more efficient (safe) one. ATMT is known to evoke off target integration of T-DNA. Pleas add PCRs confirming targeted integration of the KO cassette. It would be good practice to also confirm target integration without ectopic integration by Southern blot analysis of the mutants.
341/342: provide information on how the conidial production and germination tests were done.
Minor comments:
Most figures are in general nicely presented but in many cases captions are too small (e.g. Fig 5 c). In Figure 4 two times the same colour code was used (Vd-wt and complementation). Section 2.6 should follow after 2.2.
44: the fungus completes its life cycle
Fig 1A is not mentioned in the text.
83: Reword: Schematic of the VdTHI20 gene knock out vector design
88: and elsewhere in the manuscript: I recommend to use the wording: complemented strain instead of complementary
108: replace calculated by: measured
127: Hyphal morphology is missing in figure legend title. What medium was used for hyphal morphology microscopy? Any differences between MM (e.g. water agar) and full medium? 128 ...24 h after what? Germination?
191 Figure 6. What medium was used for the growth tests?
Author Response
Dear Reviewer:
Thank you for your comments concerning our manuscript entitled “Verticillium dahliae VdTHI20, involved in pyrimidine biosynthesis, is required for DNA repair functions and pathogenicity” (ijms-689125). Those comments are all valuable and very helpful for revising and improving our paper, as well as the important guiding significance to our researches. We have studied comments carefully and have made correction which we hope meet with approval. Revised portion are marked in red in the paper. The main corrections in the paper and the responds to the reviewer’s comments are as following:
The authors present the deletion of a gene involved in the thiamine biosynthesis pathway in Verticillium dahliae. They show that deletion of the gene VdTHI20 causes impaired vegetative growth, conidia production and germination, impaired root colonization and virulence. Especially the HIGS experiment is promising. The work is overall well done. The method section lacks some details that must be added.
Major:
Please revise the list of authors: I don´t think it should be general policy that revision of a manuscript qualifies for authorship. The same applies for a participation in material collection. In the presented work material collection does not play a major (not even a minor) role, if I am not mistaken.
Response: Thanks for the valuable suggestions. We have changed the information of author list.
110/135/149/174 and other statistics, please indicate in the legend and the graphs which two samples were compared in each students t test. (wild type vs each mutant and complemented strain)
Response: Thanks for the valuable suggestions. Sorry for these mistakes and the changes have been made accordingly.
Figure 3. Scale bars are missing.
Response: Thanks for the valuable suggestion. Sorry for this mistake and the scale bars of Figure 3 have been added.
Please connect the observed contribution to UV radiation tolerance to thiamine biosynthesis.
Response: We added the relevant content to this paper.
Thiamine was reported to play a significant role in oxidative stress response [1-5]. The orthologue of HET-P biosynthesis gene THI4, sti35 was reported to play a role in the oxidative stress response in Fusarium oxysporum [6]. VdTHI4 was reported to affect UV damage repair in S. cerevisiae and V. dahlia [7]. And also, Vd∆THI4 mutants were susceptible to oxidative stressors menadione and 2,4-DAPG. The VdΔThit mutants were also susceptible to UV damage and oxidative stressors menadione [8].
Ahn, I.P.; Kim, S.; Lee, Y.H. Vitamin B1 functions as an activator of plant disease resistance. Plant physiology 2005, 138, 1505-1515, doi:10.1104/pp.104.058693. Rapala-Kozik, M.; Kowalska, E.; Ostrowska, K. Modulation of thiamine metabolism in Zea mays seedlings under conditions of abiotic stress. Journal of experimental botany 2008, 59, 4133-4143, doi:10.1093/jxb/ern253. Rapala-Kozik, M.; Wolak, N.; Kujda, M.; Banas, A.K. The upregulation of thiamine (vitamin B1) biosynthesis in Arabidopsis thaliana seedlings under salt and osmotic stress conditions is mediated by abscisic acid at the early stages of this stress response. BMC plant biology 2012, 12, 2, doi:10.1186/1471-2229-12-2. Tunc-Ozdemir, M.; Miller, G.; Song, L.; Kim, J.; Sodek, A.; Koussevitzky, S.; Misra, A.N.; Mittler, R.; Shintani, D. Thiamin confers enhanced tolerance to oxidative stress in Arabidopsis. Plant physiology 2009, 151, 421-432, doi:10.1104/pp.109.140046. Goyer, A. Thiamine in plants: aspects of its metabolism and functions. Phytochemistry 2010, 71, 1615-1624, doi:10.1016/j.phytochem.2010.06.022. Ruiz-Roldan, C.; Puerto-Galan, L.; Roa, J.; Castro, A.; Di Pietro, A.; Roncero, M.I.; Hera, C. The Fusarium oxysporum sti35 gene functions in thiamine biosynthesis and oxidative stress response. Fungal genetics and biology : FG & B 2008, 45, 6-16, doi:10.1016/j.fgb.2007.09.003. Hoppenau, C.E.; Tran, V.-T.; Kusch, H.; Aßhauer, K.P.; Landesfeind, M.; Meinicke, P.; Popova, B.; Braus-Stromeyer, S.A.; Braus, G.H. Verticillium dahliae VdTHI4, involved in thiazole biosynthesis, stress response and DNA repair functions, is required for vascular disease induction in tomato. Environmental and Experimental Botany 2014, 108, 14-22, doi:10.1016/j.envexpbot.2013.12.015. Qi, X.; Su, X.; Guo, H.; Qi, J.; Cheng, H. VdThit, a Thiamine Transport Protein, Is Required for Pathogenicity of the Vascular Pathogen Verticillium dahliae. Molecular plant-microbe interactions : MPMI 2016, 29, 545-559, doi:10.1094/MPMI-03-16-0057-R.
Conidia formation, germination defects and cell morphology defects often are found in connection with cell wall integrity mutants. Please check sensitivity of the VdTHI20 mutants and complemented strains in comparison to wild type for sensitivity to cell-wall- perturbing agents like CFW, Congo Red to get an idea if the reduced virulence might be connected to disruption in cell wall integrity, which in turn might trigger plant defence.
Response: This is a good point. VdThit and VdTHI4 did not perform this experiment. And by the way, the time limit of major revision is only 10 days. This experiment needs about 20 days. And now our country is fighting the novel coronavirus. We are not allowed to go back to lab to do experiment right now.
Materials and Methods:
314: Please comment on the verification of the vector- was the vector sequenced or confirmed by restriction mapping?
Response: Thanks for the valuable suggestions. All the constructs were verified by enzymatic identification and DNA sequencing (Sangon, Shanghai, China).
321 ff. Give details on ATMT transformation procedure
Response: We added the details on ATMT transformation procedure in this paper.
The constructs pGn-VdTHI20 were introduced into A. tumefaciens AGL1 using electroporation. Agrobacterium tumefaciens-mediated transformation (ATMT) was used as described previously [45]. The homologous recombination transformants were selected on PDA medium (potato, 200 g/L, glucose, 20 g/L, agar, 15 g/L) with 200 μg/mL of cefotaxime, 50 μg/mL of G418 and 200 μg/mL of 5-fluoro-2′-deoxyuridine. And The complemented transformants were screened on the PDA medium with 200 μg/mL of cefotaxime and 50 μg/mL of hygromycin, in addition to 100 μg/mL geneticin for complementary transformants of VdTHI20 deletion mutants. Likewise, the plasmid pCAMgfp containing the enhanced green fluorescent protein (eGFP) and hph genes was introduced into the Vd-wt strain and the ΔVdTHI20 mutant strains using the same method. The single spore isolation was performed for all transformants, and positive transformants were verified by PCR with the test primers.
328: Gene disruption and replacement are different methods, with replacement being the more efficient (safe) one. ATMT is known to evoke off target integration of T-DNA. Pleas add PCRs confirming targeted integration of the KO cassette. It would be good practice to also confirm target integration without ectopic integration by Southern blot analysis of the mutants.
Response: Many thanks for your suggestions. The previous studies, such as AAC, VdOGDH, VdThit and VdTHI4 only did the PCR for the target integration confirmation.
341/342: provide information on how the conidial production and germination tests were done.
Response: Thanks for the valuable suggestions. This section has been improved. We have provided information on how the conidial production and germination tests.
Most figures are in general nicely presented but in many cases captions are too small (e.g. Fig 5 c). In Figure 4 two times the same colour code was used (Vd-wt and complementation). Section 2.6 should follow after 2.2.
Response: Thanks for the valuable suggestions. We are sorry for these mistakes. Changes have been made accordingly.
44: the fungus completes its life cycle
Response: “the fungus accomplishes its life cycle” was revised as “the fungus completes its life cycle”.
Fig 1A is not mentioned in the text.
Response: Thanks for the valuable suggestions. We are sorry for these mistakes. Changes have been made accordingly. Fig 1A is mentioned in section 2.1 and 4.2.
83: Reword: Schematic of the VdTHI20 gene knock out vector design
Response: Thanks for the valuable suggestions. We are sorry for these mistakes. Changes have been made accordingly.
88: and elsewhere in the manuscript: I recommend to use the wording: complemented strain instead of complementary
Response: Thanks for the valuable suggestions. Changes have been made accordingly.
108: replace calculated by: measured
Response: In this paper, “calculated” was replaced by “measured”.
127: Hyphal morphology is missing in figure legend title. What medium was used for hyphal morphology microscopy? Any differences between MM (e.g. water agar) and full medium?
Response: Thanks for the valuable suggestions. Changes have been made accordingly.
“Deletion of VdTHI20 influences conidial germination and production.” was revised as “Deletion of VdTHI20 influences hyphal morphology, conidial germination and production.”
Liquid CM broth was used for hyphal morphology microscopy
The differences between MM and CM medium are as follows:
Minimal medium (MM) agar (6 g/L NaNO3, 0.52 g/L KCl, 0.152 g/L MgSO4, 1.52 g/L KH2PO4, 0.01 g/L thiamine, 10 ml/L trace elements, 30 g/L sucrose and 15 g/L agar) ; Complete medium (CM, 6 g/L yeast extract, 6 g/L casein acid hydrolysate, 10 g/L sucrose)
128 ...24 h after what? Germination?
Response: Thanks for the valuable suggestions. We are sorry for this mistake. Hyphae of each strain were observed 24 h after germination in liquid CM broth.
191 Figure 6. What medium was used for the growth tests?
Response: Exogenous thiamine recovered the hyphal growth of ΔVdTHI20 mutants on Czapek-Dox agar plates.
We deeply appreciate you for your good comments and warm work earnestly and hope that the correction will meet with approval.
Thanks again for reviewing our manuscript.

Reviewer 2 Report
Overview:
This manuscript presents the knock down analysis of a thiamine pathway gene VdTHI20. The key findings for ∆VdTHI20 are 1) reduced mycelial growth, 2) reduced conidial germination and production caused by abnormal hyphal morphology, 3) reduced tolerance to UV damage, 4) VdTHI20 is required pathogenicity, 5) Exogenous thiamine recovered hyphal growth and 6) DsRNA of VdTHI20 confers resistance against V. dahliae in transgenic N. benthamiana lines
General comments:
Supplementary data (S1) was not available to review.
The manuscript is almost identical to a previously published 2016 paper. “Qi, X.; Su, X.; Guo, H.; Qi, J.; Cheng, H. VdThit, a Thiamine Transport Protein, Is Required for Pathogenicity of the Vascular Pathogen Verticillium dahliae. Molecular plant-microbe interactions: MPMI 2016, 29, 545-559, doi:10.1094/MPMI-03-16-0057-R.” Referenced in this paper as #10.
A VdThit (thiamine transport) gene was analysed in reference #10, and in this manuscript the authors have analysed VdTHI20 which is also thiamine related. It has been established that thiamine is crucial. The only difference between this paper and reference 10 is the gene under investigation and an extra DsRNA section. How many more genes are planned for knocked down? Why was this gene selected?
References 9 and 10 have already shown the effect of knock downs related to thiamine on pathogenicity, UV damage, growth and conidial production. Furthermore, reference 10 looked at the expression of de novo thiamine synthesis genes (VdSti35 [VDAG_01137] and Vdthi11, not mentioned in this paper?
Suggested improvements:
As the authors have used reference 10 as a template they would need to link into the previous analysis more and build on their findings.
This could be done by including the following.
A direct comparison to the relevant references (mainly reference #10), pointing out what analysis methods were repeated, what results are new and why this study was undertaken. A summary of the full pathway describing all the genes of interest VdTHI4, VdTHI20/VdTHI21, Vdthi11 etc. would be informative. This could be a figure. A summary of thiamine related genes that have previously been studied and give an overview of the results, e.g. found to have an effect on thiamine production, pathogen resistance, growth etc. This could be tabled with the references included. The authors could highlight the final section where transgenic lines expressing dsVdTHI20 showed elevated resistance to V. dahlia. Is this a novel finding?Author Response
Dear Reviewer:
Thank you for your comments concerning our manuscript entitled “Verticillium dahliae VdTHI20, involved in pyrimidine biosynthesis, is required for DNA repair functions and pathogenicity” (ijms-689125). Those comments are all valuable and very helpful for revising and improving our paper, as well as the important guiding significance to our researches. We have studied comments carefully and have made correction which we hope meet with approval. Revised portion are marked in red in the paper. The main corrections in the paper and the responds to the reviewer’s comments are as following:
Overview:
This manuscript presents the knock down analysis of a thiamine pathway gene VdTHI20. The key findings for ∆VdTHI20 are 1) reduced mycelial growth, 2) reduced conidial germination and production caused by abnormal hyphal morphology, 3) reduced tolerance to UV damage, 4) VdTHI20 is required pathogenicity, 5) Exogenous thiamine recovered hyphal growth and 6) DsRNA of VdTHI20 confers resistance against V. dahliae in transgenic N. benthamiana lines
General comments:
Supplementary data (S1) was not available to review.
Response: Thanks for the valuable suggestions. We are sorry for this mistake. We forgot to delete this section in ”ijms-template”.
The manuscript is almost identical to a previously published 2016 paper. “Qi, X.; Su, X.; Guo, H.; Qi, J.; Cheng, H. VdThit, a Thiamine Transport Protein, Is Required for Pathogenicity of the Vascular Pathogen Verticillium dahliae. Molecular plant-microbe interactions: MPMI 2016, 29, 545-559, doi:10.1094/MPMI-03-16-0057-R.” Referenced in this paper as #10.
A VdThit (thiamine transport) gene was analysed in reference #10, and in this manuscript the authors have analysed VdTHI20 which is also thiamine related. It has been established that thiamine is crucial. The only difference between this paper and reference 10 is the gene under investigation and an extra DsRNA section. How many more genes are planned for knocked down? Why was this gene selected?
Response: We planned toknock down THI20 (HMP-PP synthesis and thiamine degradation) and THI80 (Thiamin pyrophosphokinase), while VdTHI20 homologous gene OsTHIC was reported to be involved in stress response in rice. Thus we first did the knock down of THI20.
References 9 and 10 have already shown the effect of knock downs related to thiamine on pathogenicity, UV damage, growth and conidial production. Furthermore, reference 10 looked at the expression of de novo thiamine synthesis genes (VdSti35 [VDAG_01137] and Vdthi11, not mentioned in this paper?
Response: Thanks for the valuable suggestions. According to your suggestions, we have added the mention about references 9 and 10 have already shown the effect of knock downs related to thiamine on pathogenicity, UV damage, growth and conidial production.
The expression of VdSti35 [VDAG_01137] and Vdthi11 is a good suggestion. However, we are not allowed to go back to school due to the novel coronavirus in China.
Suggested improvements:
As the authors have used reference 10 as a template they would need to link into the previous analysis more and build on their findings.
This could be done by including the following.
A direct comparison to the relevant references (mainly reference #10), pointing out what analysis methods were repeated, what results are new and why this study was undertaken. A summary of the full pathway describing all the genes of interest VdTHI4, VdTHI20/VdTHI21, Vdthi11 etc. would be informative. This could be a figure. A summary of thiamine related genes that have previously been studied and give an overview of the results, e.g. found to have an effect on thiamine production, pathogen resistance, growth etc. This could be tabled with the references included. The authors could highlight the final section where transgenic lines expressing dsVdTHI20 showed elevated resistance to V. dahlia. Is this a novel finding?
Response: The relative content was added to the introduction part of paper.
Thiamine consists of two aromatic components, a pyrimidine (2-methyl-4-amino-5-hydroxymethylpyrimidine, HMP) and a thiazole (4-methyl-5-β-hydroxyethylthiazole, HET), which are synthesized in independent branches of the thiamine synthesis pathway. Thiamine pyrophosphate, phosphorylated thiamine (TPP), is the predominant active form of thiamine [8,9]. In Saccharomyces cerevisiae, many genes were found to be involved in thiamine pathway. THI5, THI11, THI12 and THI13 are THI5 gene family members that their function is redundant in HMP-P synthesis [10]. Three members of the THI20 family proteins, THI20/21/22, were characterized in S. cerevisiae [11]. THI20 and THI21 are two redundant genes that encode the HMP-P kinase of S. cerevisiae. THI20/THI21 can not only phosphorylate HMP to HMP-P but also phosphorylate HMP-P to HMP-PP [12]. Moreover, THI20 has been indicated to be a trifunctional protein with both thiamine biosynthetic and degradative activity [13]. THI4, involved in HET-P synthesis in S. cerevisiae, could play a role in DNA damage tolerance [14]. THI6 has been showed to have the thiamin-phosphate pyrophosphorylase and hydroxyethylthiazole kinase activity [15]. THI80 is a thiamin-pyrophosphokinase gene that catalyzes thiamine to the active form (TPP) [16]. THI10, which encodes a thiamine transport protein that transports the thiamine from periplasm to the cell [17]. PHO3 is a periplasmic acid phosphatase that allows thiamin phosphates hydrolysis in periplasm [18]. In recent years, several genes in the thiamine biosynthesis pathway have been found in V. dahliae. VdTHI4, involved in HET-P synthesis, is required for fungal induction in tomato, as demonstrated by the infection analysis of the knockout strain ∆VdTHI4 [19]. VdThit, a thiamine transport protein, is important for the pathogenicity of V. dahliae, as supported by infection assays with the V. dahliae VdΔTHit deletion strain (Table 1)[20].HET biosynthesis and thiamine transport for the thiamine biosynthesis pathway seem to influence the virulence and pathogenesis of V. dahlia. Only one THI20 family gene, VdTHI20, was found in V. dahliae (Fig. 1). However, the mechanism by which VdTHI20 contributes to the virulence and pathogenesis of V. dahliae has not been reported.
Table 1. Genes involved in thiamine biosynthesis.
|
Gene |
Protein function |
References |
|
THI4 |
HET-P synthesis; Stress response and DNA repair |
[14,19] |
|
THI5/THI11/THI12/THI13 |
HMP-P synthesis |
[10] |
|
THI6 |
thiamin-phosphate pyrophosphorylase and hydroxyethylthiazole kinase |
[15] |
|
THI10 |
Thiamine transport protein |
[17] |
|
THI20 |
HMP-PP synthesis and thiamine degradation |
[11,12,21] |
|
THI21/THI22 |
HMP-PP synthesis |
[11,12] |
|
THI80 |
Thiamin pyrophosphokinase |
[16] |
|
VdThit |
Thiamine transport protein; Vegetable growth, reproduction and invasive hyphal growth |
[20] |
|
PHO3 |
Thiamin phosphates hydrolysis |
[18] |
Figure 1. A schematic of the thiamine biosynthesis pathway. Black italics represent the genes found in S. cerevisiae. Green italics represent the genes found in V. dahliae. Red italics indicate genes found in both S. cerevisiae and V. dahliae. Blue oval represents thiamine transport protein.
Special thanks to you for your good comments. We appreciate for Reviewers’ warm work earnestly and hope that the correction will meet with approval.

Round 2
Reviewer 1 Report
Dear Authors,
thank you for improving the manuscript. I understand that it is difficult for you to conduct any experiments at the moment due to the corona virus outbreak.
For the graphs I still would recommend to increase not the letters (A,B,C,D etc of subimages/graphs) but the graph legend. For the statistics in the graphs I recommend sth like the following example:
Author Response
Dear Reviewer:
Thank you for your comments concerning our manuscript entitled “Verticillium dahliae VdTHI20, involved in pyrimidine biosynthesis, is required for DNA repair functions and pathogenicity” (ijms-689125). Those comments are all valuable and very helpful for revising and improving our paper, as well as the important guiding significance to our researches. We have studied comments carefully and have made correction which we hope meet with approval. Revised portion are marked in red in the paper. The main corrections in the paper and the responds to the reviewer’s comments are as following:
thank you for improving the manuscript. I understand that it is difficult for you to conduct any experiments at the moment due to the corona virus outbreak.
For the graphs I still would recommend to increase not the letters (A,B,C,D etc of subimages/graphs) but the graph legend. For the statistics in the graphs I recommend sth like the following example:
Response: Thanks for the valuable suggestions. We have increased the graph legend and improved the statistics accordingly.
We deeply appreciate you for your good comments and warm work earnestly and hope that the correction will meet with approval.
Thanks again for reviewing our manuscript.
Reviewer 2 Report
The revised manuscript has improved and is much more transparent.
I previously requested the authors to make the Supplementary data (S1) available for review.
The authors have responded by deleting the link to the supplementary data?
Can the authors please include the supplementary data (S1)?
Author Response
Dear Reviewer:
Thank you for your comments concerning our manuscript entitled “Verticillium dahliae VdTHI20, involved in pyrimidine biosynthesis, is required for DNA repair functions and pathogenicity” (ijms-689125). Those comments are all valuable and very helpful for revising and improving our paper, as well as the important guiding significance to our researches. We have studied comments carefully and have made correction which we hope meet with approval. Revised portion are marked in red in the paper. The main corrections in the paper and the responds to the reviewer’s comments are as following:
The revised manuscript has improved and is much more transparent.
I previously requested the authors to make the Supplementary data (S1) available for review.
The authors have responded by deleting the link to the supplementary data?
Can the authors please include the supplementary data (S1)?
Response: Thanks for the valuable suggestions. Sorry that we didn’t get ourselves understood. Actually, we do not have the supplementary data (S1) in our manuscript. We used the ”ijms-template” for adjusting the format of our manuscript. The ”ijms-template” has the supplementary data (S1). We forgot to delete the supplementary data (S1) in ”ijms-template” when we uploaded our manuscript. Sorry again about this confusion.
We deeply appreciate you for your good comments and warm work earnestly and hope that the correction will meet with approval.
Thanks again for reviewing our manuscript.